# 3D Deployment Optimization of Wireless Sensor Networks for Heterogeneous Functional Nodes

**DOI:** 10.3390/s25051366

**Published:** 2025-02-23

**Authors:** Zean Lu, Chengqun Wang, Peng Wang, Weiqiang Xu

**Affiliations:** 1School of Information Science and Engineering, Zhejiang Sci-Tech University, Hangzhou 310018, China; 202230603089@mails.zstu.edu.cn; 2Key Laboratory of Intelligent Textile and Flexible Interconnection of Zhejiang Province, Zhejiang Sci-Tech University, Hangzhou 310018, China; 2023220603067@mails.zstu.edu.cn (P.W.); wqxu@zstu.edu.cn (W.X.)

**Keywords:** deployment optimization of heterogeneous wireless sensor networks, K-coverage, C-connectivity, improved bird optimization algorithm, minimum spanning tree algorithm

## Abstract

The optimization of wireless sensor network (WSN) deployment is a current research hotspot, particularly significant in industrial applications. While some existing optimization methods focus more on balancing network coverage, connectivity, and deployment costs, aligning them with practical needs compared to single-performance optimization schemes, they still tend to be overly idealized. In practical applications, networks often face monitoring requirements for different data types, and some single-function sensors can be integrated into multifunctional sensors capable of monitoring multiple types of data. When encountering diverse data detection needs in a target area, this integration can be further considered to reduce deployment costs. Therefore, this paper designs a new multi-objective optimization problem aimed at optimizing heterogeneous-function wireless sensor networks, balancing coverage, connectivity, and cost, while introducing an additional cost dimension to meet the monitoring needs of different functional sensors in specific areas. This problem is a typical non-convex, multimodal, NP-hard problem. To address this, an improved Secretary Bird Optimization Algorithm (ISBOA) is proposed, incorporating Gaussian Cuckoo Mutation and a smooth exploitation mechanism. The algorithm is compared with the original SBOA, Particle Swarm Optimization (PSO), Whale Optimization Algorithm (WOA), and Northern Goshawk Optimization (NGO). Simulation results demonstrate that ISBOA exhibits a faster convergence speed and higher accuracy in both the 23 benchmark functions and the newly designed multi-objective optimization problem, significantly overcoming the shortcomings of the compared algorithms. Finally, for large-scale optimization problems, a minimum spanning tree domain reduction strategy is proposed, which significantly improves solving efficiency with a moderate sacrifice in accuracy.

## 1. Introduction

### 1.1. Background

Wireless Sensor Networks (WSNs) consist of a large number of sensor nodes that form a self-organizing network structure that is widely used in various fields, such as environmental monitoring, intelligent transportation, military reconnaissance, and health monitoring [1,2]. With the rapid development of the Internet of Things (IoT) and the gradual implementation of future factories, the WSN application scenarios are continuously expanding. For example, manufacturing industries such as textiles and pharmaceuticals require the real-time monitoring of key parameters, such as temperature, humidity, and pressure, in the working environment during production to ensure that product quality meets stringent standards. Without precise control over these basic environmental data, the produced items may not meet usage requirements. Therefore, the rational deployment of sensor node layouts, enabling WSN to possess excellent network performance, has become a significant driver in promoting the intelligent development of the manufacturing sector.

The core objectives of network performance optimization include precise coverage, reliable communication performance, and controllable deployment costs. However, past research has predominantly focused on enhancing the coverage performance of WSN, overlooking multifaceted practical needs. In recent years, an increasing number of studies have recognized the limitations of single-objective optimization and have gradually incorporated more factors to consider, such as adding comprehensive considerations of network connectivity and the number of sensor nodes, while optimizing network coverage. Although this approach more accurately reflects the actual requirements, it still does not fully capture the complexity of WSNs in real-world deployments. Therefore, we have improved the deployment of WSNs based on the latest multi-objective optimization methods. Especially in actual monitoring environments, where the types of data transmitted are diverse, we have chosen to deploy heterogeneous sensor node networks. The term “heterogeneous” not only refers to the functional differences of sensor nodes but also includes differences in coverage and communication ranges. Moreover, some heterogeneous nodes are capable of monitoring multiple types of data, which requires us to consider various factors from multiple dimensions during the optimization process. This multi-dimensional optimization approach not only better aligns with practical application needs, but it also represents the future trend in the optimization of WSN deployment.

### 1.2. Related Works

Optimization of WSN deployment has always been a hot topic in research. As early as 1998, the study by Gupta, P. and Kumar, P. R. discussed the issues of node deployment and network connectivity [3]. Although they did not directly study the optimized deployment of WSNs, their work laid the theoretical foundation for many subsequent deployment optimization algorithms. In 2001, Meguerdichian et al. first introduced the coverage problem of wireless sensor networks, developed a coverage model, and focused on how to achieve maximum coverage through the reasonable deployment of sensor nodes, thus laying the foundation for WSN coverage optimization [4].

Early studies mainly focused on the optimization of a single objective, such as coverage or connectivity, to measure the performance of the network. Usually, a two-dimensional target monitoring area is known, a certain number of sensor nodes are given, and the sensor nodes are deployed in the monitoring area to obtain the best network coverage performance. However, in actual situations, the optimization of a single objective can neither represent the overall performance of WSN nor meet actual needs. Therefore, in 2008, Ghosh and Das systematically classified and deeply discussed coverage and connectivity models, with a particular emphasis on achieving maximum coverage and connectivity for the network by optimizing the deployment positions of sensor node groups [5]. Although such improvements provided more comprehensive considerations for research, there remains a certain gap in practical applications. Generally, the coverage and connectivity performance of wireless sensor networks will improve with an increase in the number of sensor nodes and an increase in the coverage radius of the sensor nodes themselves. In other words, some key factors, such as the number of sensor nodes, the radius of coverage, and the location of the sensor nodes in the target monitoring area, determine the quality of the coverage and connectivity performance of the wireless sensor network. However, improving the overall performance of the wireless sensor network in this way will increase the deployment cost. Therefore, we must consider the impact of both on the overall network performance at the same time. Subsequently, Chen et al. proposed a multi-objective optimization method to achieve a balance between cost and reliability in 2022, which further improved the complete efficiency and effect of WSN deployment optimization [6]. Overall, research on the deployment optimization of wireless sensor networks has gradually shifted from single-objective optimization to multi-objective optimization. It requires the network to cover all target areas as much as possible, maintain good communication performance, ensure the stability of the entire network, and also consider the overall cost of deployment of sensor nodes. By deploying a certain number of sensor nodes, it aims to achieve better network performance [7,8], seeking a balance between coverage, connectivity, and deployment cost in WSNs.

At present, most of the research on the optimal deployment of wireless sensor networks is based on two-dimensional considerations. In two-dimensional deployment, the coverage and perception range of sensor nodes is limited, especially in environments with varying heights or non-flat terrain. Some examples include field monitoring, building structure monitoring, or underground resource detection. When sensor nodes are deployed in these scenarios, they must be able to cover or monitor areas at different heights or depths [9,10]. At the same time, in some scenarios, three-dimensional deployment can more effectively save energy and optimize costs. For example, through reasonable vertical deployment, the number of nodes or the communication distance between nodes can be reduced, thereby reducing energy consumption and maintenance costs. Therefore, only by considering deployment in three-dimensional space can we more realistically simulate and meet the actual application needs [11].

In practical application scenarios, it is necessary to consider that different target locations within the monitoring area may require different types of data feedback. This means that the corresponding functional sensors should be deployed around each target point in the target area to monitor the respective data. In addition, some targets require the transmission of more than one type of data. Therefore, such target points will only be considered as valid coverage when all the necessary sensors are present and fully cover the area. If any type of sensor does not cover the target point, the data collected by the gateway will be incomplete, which will directly affect the entire wireless sensor network. Furthermore, on the basis of practical needs, some single-function sensors can be integrated. The integrated sensors have not only all the functions of the sensors being integrated, but also come at a lower cost. Considering these characteristics, we need to further contemplate the overall deployment cost of the sensor network. A wireless sensor network composed of different types of sensor nodes such as this can also be called a heterogeneous functional wireless sensor network.

Therefore, this paper proposes a novel multi-objective optimization problem aimed at achieving a balance between overall network performance and cost while satisfying the monitoring requirements of specific regions. Since this problem belongs to NP-hard problems, the solution relies on swarm intelligence optimization algorithms. By using swarm intelligence optimization algorithms, sensor nodes can be deployed in better locations, leading to improved network performance. For algorithm selection, we adopted the SBOA proposed by Fu Youfa et al. in 2024 [12]. This algorithm is inspired by the survival behavior of the secretary bird. In the exploration phase, the algorithm simulates the hunting process of the secretary bird by combining differential evolution, Brownian motion, and Lévy flight strategies. This not only enhances global exploration capability and local adjustment ability, but also improves the diversity of initial solutions, coverage efficiency, and optimization accuracy, thus achieving precise sensor layout optimization. In the development phase, SBOA simulates the escape behavior of the secretary bird from predators and uses two escape strategies. The first strategy combines random Brownian motion and a time factor to gradually reduce the disturbance range and optimize the distribution of sensor nodes and network performance. The second strategy generates new positions, using normal distribution random numbers and random solutions to expand the search range. The equal probability application of these two strategies effectively coordinates and enhances the multi-objective optimization of the wireless sensor network. The two phases alternate iteratively until the termination condition is met, eventually finding the optimal solution to the optimization problem.

To further improve the convergence and accuracy of the SBOA in the newly designed multi-objective optimization problem, we made two improvements to the algorithm. First, we added the Gaussian cuckoo mutation before the exploration phase [13]; second, we added a smoothing development mechanism after the development phase [14]. The Gaussian cuckoo mutation combines the Gaussian mutation mechanism and cuckoo search by applying random numbers, following a normal distribution to the original position vector, to generate new positions. Together with the cuckoo bird parasitic breeding strategy, it achieves an effective combination of global and local search, thus improving optimization accuracy. The smoothing development mechanism consists of unordered dimension sampling, random crossover, and sequential mutation. Unordered dimension sampling combines random and static dimensions to effectively reduce dimensions when approaching the optimal solution, preventing population sparsity and optimizing computation time. Random crossover determines the exploration radius based on the distance between individuals, enhancing exploration ability and ensuring that a large radius escapes local optima and a small radius enters the exploitation phase. Sequential mutation explores the space between individuals based on index adjacency, effectively solving the dimension disaster problem and preventing the rapid contraction of the search space. These two improvements complement each other, significantly enhancing the exploration effect and avoiding the computational issues of harmonic and geometric means by using arithmetic averages.

Furthermore, we chose PSO [15], WOA [16], and NGO [17] as the comparative algorithms for ISBOA. PSO, proposed by James Kennedy and Russell C. Eberhart in 1995, is inspired by bird flocking behavior and is simple and efficient, widely used in various optimization tasks [18]. WOA, proposed by Seyedali Mirjalili et al. in 2016, is inspired by the bubble net hunting method of humpback whales. It balances global and local searches well, and it is suitable for high-dimensional optimization problems [19]. NGO, proposed by Mohammad Dehghani et al. in 2021, simulates the recognition, chasing, and escape behaviors of the northern goshawk during hunting. It has strong global search capability, fast convergence, and simple parameter settings, making it particularly suitable for high-dimensional, multi-peak, and multi-constraint optimization tasks [20]. These three algorithms, in terms of execution time and characteristics, are highly representative and provide good comparative choices for solving this problem. Finally, through numerous experiments, we verified that ISBOA demonstrates faster convergence speed and higher search accuracy compared to other algorithms in this multi-objective optimization problem.

In practical applications, the multi-objective optimization problem we propose typically involves large-scale networks and a large number of variables. This often requires swarm intelligence algorithms to spend considerable amounts of time and computational resources in order to initialize feasible solutions that meet the various constraints. Given the complexity of these tasks, a more efficient approach is needed to reduce the time spent on initialization. To address this challenge, we introduce an initialization strategy that leverages the minimum spanning tree (MST) algorithm [21]. This strategy first calculates the shortest connected path between the target points and the randomly placed sensors, treating this path as a fitness function to guide the optimization process. Following this, the algorithm refines the sensor positions by optimizing this path, ultimately reducing the overall distance. By initializing the optimal solution around the MST, we effectively reduce the feasible region for sensor deployment, which leads to the faster satisfaction of the deployment constraints during the initialization phase. While this method results in a significant reduction in solution time, it may come at the cost of a slight loss in accuracy. However, the trade-off remains acceptable, and the strategy proves to be highly practical for solving large-scale optimization problems. Its efficiency offers considerable practical value, especially in scenarios where rapid deployment is crucial.

To clearly illustrate the differences between our research and the existing related work, we have included a comparison table in this section, as shown below. Table 1 summarizes the main features of different studies and highlights their differences to the approach we propose.

### 1.3. Contributions

The main contributions of this paper are as follows:Introducing an additional cost dimension: This paper proposes a new multi-objective optimization problem aimed at optimizing heterogeneous wireless sensor networks, balancing coverage, connectivity, and cost, while introducing an additional cost dimension to achieve optimization that is more aligned with practical applications.Considering monitoring requirements for specific regions: The paper also addresses the typical challenges in real-world applications, where networks must meet various types of data monitoring demands. Specifically, it explores how to use single-function sensors that can be integrated into multi-functional sensors capable of monitoring multiple types of data, thus reducing deployment costs while meeting the monitoring needs of different functional sensors in targeted areas.Proposing the ISBOA and performing simulation validation: This paper presents an improved Serpent Eagle Optimization Algorithm (ISBOA), which integrates a Gaussian cuckoo mutation and a smooth development mechanism. Compared with SBOA, PSO, WOA, and NGO, the ISBOA demonstrates significant performance improvements. Simulation results show that the ISBOA achieves faster convergence and higher precision across 23 benchmark functions and the newly designed multi-objective optimization problem.Proposing a minimum spanning tree domain reduction strategy: For large-scale optimization problems, this paper introduces a minimum spanning tree domain reduction strategy, significantly improving solution efficiency while sacrificing some accuracy.

### 1.4. Organization

The structure of the remainder of the paper is as follows. Section 2 describes the multi-objective optimization problem and provides a formal formulation. Section 3 introduces the methodology and the improvements made. Section 4 presents the discussion and evaluation of the results. Finally, Section 5 concludes the paper and outlines directions for future improvements.

## 2. Multi-Objective Optimization Problem Formulation

In WSN deployment, the coverage, connectivity, and deployment cost are three interrelated core objectives. Coverage determines the sensor’s ability to monitor the target area, connectivity ensures data can be transmitted effectively between nodes, and deployment cost limits the number of nodes and resource usage. High coverage and connectivity typically require more nodes or larger sensing and communication ranges, but this increases the cost; conversely, reducing cost may affect coverage and connectivity performance. Therefore, multi-objective optimization algorithms balance these three factors to find the optimal deployment plan, minimizing costs while meeting monitoring requirements. Section 2.1, Section 2.2 and Section 2.3 provide detailed designs and discussions on WSN coverage performance, connectivity, and deployment cost, respectively, while Section 2.4 summarizes these three aspects, balancing their relationships and setting relevant constraints to ensure compliance with practical application needs.

In the three-dimensional space, let *D* be the region that needs to be monitored, with the length, width, and height denoted as $M_{a}$, $M_{b}$, and $M_{c}$, respectively. This region is discretized into grid points with an interval of $S_{c}$, and the grid points are evenly distributed. The set of all discrete grid points within space *D* represents the potential locations where sensor nodes are to placed. We use the parameter $S_{c}$ to control the discretization level of the grid points in the monitoring space *D*. As $S_{c}$ approaches 0, these discrete grid points can more accurately represent the locations within space *D*. We denote the set of potential sensor node locations as $H=\{h_{1},h_{2},h_{3},\ldots ,h_{s}\}$, where *s* is the total number of potential placement locations. Therefore, the position of $h_{i}$ in the three-dimensional space is represented as $\left(X_{h_{i}},Y_{h_{i}},Z_{h_{i}}\right)$, as shown in Figure 1.

We define the upper limit for the number of sensor nodes as *n*, where n≤s. The sensor types are divided into two categories as follows: single-function sensors, with *K* types; and multi-functional sensors, which are integrated from single-function sensors, with *N* types. Thus, we define the sensor type set as $V_{\mathrm{all}}=\left\{v_{1},v_{2},\ldots ,v_{K},v_{K+1}\left(i+j\right),\ldots ,v_{K+N}\left(p+q\right)\right\}$, where $v_{K+1}\left(i+j\right),v_{K+N}\left(p+q\right)$ represents integrated sensors formed by combining two or more single-function sensors. For example, the integrated sensor $v_{K+1}\left(i+j\right)$ is formed by integrating sensor types $v_{i}$ and $v_{j}$, resulting in a lower production cost and an economic benefit. Each sensor type has an associated production cost, denoted as $S_{\mathrm{all}}=\left\{s_{1},s_{2},\ldots ,s_{K},s_{K+1}\left(i+j\right),\ldots ,s_{K+N}\left(p+q\right)\right\}$, where the values in $S_{\mathrm{all}}$ correspond to the sensor types in $V_{\mathrm{all}}$. For instance, the price of sensor type $v_{K}$ is $s_{K}$, and the price of the integrated sensor $v_{K+N}\left(p+q\right)$ is $s_{K+N}\left(p+q\right)$.

At the same time, we define the set of target points as $T=\{t_{1},t_{2},\ldots ,t_{m}\}$, where *m* is the total number of target points and m≤s. We use $X_{h_{j}}^{v_{i}}=1$ to indicate that a sensor node of type $v_{i}$ is located at position $h_{j}$, and 0 if otherwise. Heterogeneous sensor nodes have different sensing radii and communication radii, and the communication radius of the same type of sensor is always larger than the coverage radius. We use $R_{\mathrm{cov}}^{v}$ to represent the coverage radius and $R_{\mathrm{com}}^{v}$ to represent the communication radius, as shown in Figure 2a.

### 2.1. Coverage Model

In an actual wireless sensor network, the sensing range of sensor nodes attenuates as the distance increases [22,23]. In a monitoring space *D*, the probability that a target point $t_{j}$ is detected by a sensor of type $v_{k}$, located at position $h_{i}\in H$, is given by the following:(1)$$\phi \left(h_{i},t_{j},v_{k}\right)=\left\{\begin{array}{cc} 1, & \mathrm{if}\;d\left(h_{i},t_{j}\right)\leq R_{\mathrm{cov}}^{v_{k}}-R_{e}\left(\mathrm{Cov}\right)\\ e^{-\alpha d(h_{i},t_{j})}, & \mathrm{if}\;|d\left(h_{i},t_{j}\right)-R_{\mathrm{cov}}^{v_{k}}|<R_{e}\left(\mathrm{Cov}\right)\\ 0, & \mathrm{otherwise} \end{array}\right.$$
where α is the probability parameter and $R_{e}\left(\mathrm{Cov}\right)$ represents the uncertainty measure in sensor monitoring, as shown in Figure 2b. The Euclidean distance between the position $h_{i}$ and the target point $t_{j}$ is given by the following:(2)$$d\left(h_{i},t_{j}\right)=\sqrt{{\left(x_{h_{i}}-x_{t_{j}}\right)}^{2}+{\left(y_{h_{i}}-y_{t_{j}}\right)}^{2}+{\left(z_{h_{i}}-z_{t_{j}}\right)}^{2}}$$

Additionally, a threshold $\mu_{1}$ in the range [0,1] is introduced. If $\phi \left(h_{i},t_{j},v_{k}\right)>\mu_{1}$, the target point $t_{j}$ is considered detectable by the sensor at $h_{i}$ of type $v_{k}$; otherwise, it is not.

The coverage function is defined as follows:(3)$$\mathrm{Cov}\left(h_{i},t_{j},v_{k}\right)=\left\{\begin{array}{cc} 1, & \mathrm{if}\;\phi \left(h_{i},t_{j},v_{k}\right)>\mu_{1}\\ 0, & \mathrm{otherwise} \end{array}\right.$$

In the detection area, the number of sensors that cover the target point $t_{j}$ is defined as $P_{\mathrm{cov}}\left(t_{j}\right)$, and it is calculated as follows:(4)$$P_{\mathrm{cov}}\left(t_{j}\right)=\sum_{h_{i}\in H}\sum_{v_{k}\in V_{\mathrm{all}}}\mathrm{Cov}\left(h_{i},t_{j},v_{k}\right)\cdot X_{h_{i}}^{v_{k}}$$

Thus, the first objective function, coverage, is given by the following:(5)$$f_{1}=\frac{\sum_{t_{j}\in T}P_{\mathrm{cov}}\left(t_{j}\right)}{m}$$

### 2.2. Communication Model

We adopt a probabilistic communication model [24]. Suppose that two sensors have the same communication range and are located at positions $h_{i}$ and $h_{j}$ in the search space, with i≠j. The communication probability between these two sensors is given by the following:(6)$$\partial \left(h_{i},h_{j}\right)=\left\{\begin{array}{cc} 1, & \mathrm{if}\;d\left(h_{i},h_{j}\right)\leq R_{\mathrm{com}}^{v}-R_{e}\left(\mathrm{Com}\right)\\ e^{-\theta_{1}{\left[d\left(h_{i},h_{j}\right)-\left(R_{\mathrm{com}}^{v}-R_{e}\left(\mathrm{Com}\right)\right)\right]}^{\theta_{2}}}, & \mathrm{if}\;|d\left(h_{i},h_{j}\right)-R_{\mathrm{com}}^{v}|<R_{e}\left(\mathrm{Com}\right)\\ 0, & \mathrm{otherwise} \end{array}\right.$$
where $R_{e}\left(\mathrm{Com}\right)$ represents the uncertainty measure in the sensor communication range, as shown in Figure 2c, and $\theta_{1}$ and $\theta_{2}$ are the probability parameters. The Euclidean distance between $h_{i}$ and $h_{j}$ is denoted by $d(h_{i},h_{j})$.

A threshold $\mu_{2}$ in the range [0,1] is introduced. If $\partial \left(h_{i},h_{j}\right)>\mu_{2}$ and the sensors are located at $h_{i}$ and $h_{j}$, they can communicate; otherwise, they cannot. This is represented by the following:(7)$$\mathrm{Com}\left(h_{i},h_{j}\right)=\left\{\begin{array}{cc} 1, & \mathrm{if}\;\partial \left(h_{i},h_{j}\right)>\mu_{2}\\ 0, & \mathrm{otherwise} \end{array}\right.$$

We define $P_{\mathrm{com}}\left(h_{i}\right)$ as the number of sensor nodes that can communicate with the sensor at position $h_{i}$, which is calculated as follows:(8)$$P_{\mathrm{com}}\left(h_{i}\right)=\sum_{h_{j}\in H\setminus h_{i}}\sum_{v_{k}\in V_{\mathrm{all}}}\sum_{v_{p}\in V_{\mathrm{all}}}\mathrm{Com}\left(h_{i},h_{j}\right)\cdot X_{h_{i}}^{v_{k}}\cdot X_{h_{j}}^{v_{p}}$$

Finally, the communication performance of the sensor nodes is represented by $f_{2}$, expressed as follows:(9)$$f_{2}=\frac{\sum_{h_{i}\in H}P_{\mathrm{com}}\left(h_{i}\right)}{\sum_{v_{k}\in V_{\mathrm{all}}}\sum_{h_{i}\in H}X_{h_{i}}^{v_{k}}}$$

### 2.3. Cost Model

Let the set of sensor types be denoted as $V_{\mathrm{all}}=\left\{v_{1},v_{2},\ldots ,v_{K},v_{K+1(i+j)},\ldots ,v_{K+N(p+q)}\right\}$. We divide this set into two parts based on the type and functionality of the sensors. One part consists of sensors with a single function, denoted as $V_{\mathrm{NI}}=\left\{v_{1},v_{2}\ldots v_{j},v_{k}\ldots v_{p},v_{q}\right\}$. This set includes sensors like $v_{i}$ and $v_{j}$, which can be integrated into multifunctional sensors such as $v_{1}\left(i+j\right)$, as well as sensors like $v_{1},\;v_{2},\;v_{3}$, which have no integration attributes. The other part consists of integrated sensor types, such as $v_{1}\left(i+j\right)$ and $v_{N}\left(p+q\right)$, which are represented by the set $V_{I}=\left\{v_{1}\left(i+j\right),\ldots ,v_{N}\left(p+q\right)\right\}$.

Similarly, we divide the total price set $S_{\mathrm{all}}=\left\{s_{1},s_{2},s_{3},\ldots ,s_{K},s_{K+1(i+j)},\ldots ,s_{K+N(p+q)}\right\}$ into two parts based on the following sensor types: $S_{\mathrm{NI}}=\left\{s_{1},s_{2},s_{3},\ldots ,s_{i},s_{j},s_{k},\ldots ,s_{p},s_{q}\right\}$ and $S_{I}=\left\{s_{1}\left(i+j\right),\ldots ,s_{N}\left(p+q\right)\right\}$.

The cost of the wireless sensor network is determined by the number of sensor nodes and the manufacturing cost of the sensors themselves. This is expressed by the objective function $f_{3}$, expressed as follows:(10)$$f_{3}=\sum_{h_{i}\in H}\sum_{v_{k}\in V_{\mathrm{NI}}}\sum_{s_{k}\in S_{\mathrm{NI}}}X_{h_{i}}^{v_{k}}s_{k}+\sum_{h_{i}\in H}\sum_{v_{k}\in V_{I}}\sum_{s_{k}\in S_{I}}X_{h_{i}}^{v_{k}}s_{k}=\sum_{h_{i}\in H}\sum_{v_{k}\in V_{\mathrm{all}}}\sum_{s_{k}\in S_{\mathrm{all}}}X_{h_{i}}^{v_{k}}s_{k}$$

### 2.4. Problem Summary

The multi-objective optimization problem in WSN focuses on the following three key objectives: the coverage, connectivity, and deployment cost of sensor nodes. By balancing these objectives, we aim to determine the optimal deployment positions. Furthermore, practical application requirements impose several constraints on the WSN [6], which are summarized as follows:$$\min_{X}\left\{-f_{1},-f_{2},f_{3}\right\}$$(11)$$\begin{array}{cc} s.t.\;\; & C_{1}:G_{\mathrm{WSN}}=1,\\ \; & C_{2}:P_{\mathrm{cov}}\left(t_{j}\right)\geq K,\;\forall j,\\ \; & C_{3}:P_{\mathrm{com}}\left(h_{j}\right)\geq C,\;\forall j,\\ \; & C_{4}:X_{h_{j}}^{v_{i}}\in \left\{0,1\right\},\;\forall i,j,\\ \; & C_{5}:\sum_{i\in V_{\mathrm{all}}}X_{h_{j}}^{v_{i}}\leq 1,\;\forall j,\\ \; & C_{6}:\min_{k}\left\{\sum_{h_{i}\in H}\mathrm{Cov}\left(h_{i},t_{j},v_{k}\right)\right\}+\sum_{h_{i}\in H}\mathrm{Cov}\left(h_{i},t_{j},v_{n}\left(a+b\right)\right)\geq K,\;k\in \left\{a,b\right\}. \end{array}$$
where $X_{h_{j}}^{v_{i}}$ is the optimization variable; $C_{1}$ ensures that the wireless sensor network is fully connected; $C_{2}$ ensures that each target point in the network is covered by at least *K* sensor nodes; $C_{3}$ ensures the *C*-connectivity between sensor nodes; $C_{4}$ indicates that the optimization variables can only take values 0 or 1; $C_{5}$ ensures that only one sensor node can be placed at each position in the network; and $C_{6}$ ensures that if a target point $t_{j}$ requires both $v_{a}$ and $v_{b}$ types of sensor nodes for monitoring, and $v_{a}$ and $v_{b}$ can be integrated into $v_{n}\left(a+b\right)$, then the target point must be covered at least *K* times.

## 3. Solution

The multi-objective optimization problem is NP-hard, making swarm intelligence algorithms essential for effective solutions. This paper proposes the ISBOA, an improved version of the SBOA, integrating Gaussian cuckoo mutation and smooth development mechanisms to optimize wireless sensor network deployment. To address the challenge of long computation times for large-scale problems, a minimum spanning tree domain reduction strategy is introduced, improving efficiency while moderately sacrificing accuracy.

### 3.1. ISBOA Algorithm Implementation

The ISBOA is an improvement of the SBOA, which is inspired by the survival behavior of the secretary bird. In the exploration phase, it simulates the hunting process of the secretary bird, enhancing global exploration and local adjustment capabilities by combining differential evolution, Brownian motion, and Levy flight strategies. This improves the diversity of solutions, coverage efficiency, and optimization accuracy, thus optimizing sensor layout. In the development phase, it simulates the secretary bird’s behavior of evading predators, using two escape strategies as follows: one combines random Brownian motion and time factors to optimize node distribution, while the other expands the search range by using normal distribution and random solutions. These two strategies work together to improve the multi-objective optimization performance of the wireless sensor network.

In addition, we have integrated strategies such as Gaussian mutation, cuckoo search, and smooth development mechanism into the SBOA to further enhance the optimization performance of the wireless sensor network. In terms of global search, Gaussian mutation and cuckoo search improve population diversity, escape from local optima, and global search efficiency. Sequential dimension sampling and random crossover expand the search space, ensuring global exploration and improving algorithm robustness. For local search, sequential mutation optimizes solution precision and ensures stable convergence. Overall, these strategies effectively enhance sensor node layout optimization, improving network coverage, connectivity, and communication efficiency, while adapting to complex dynamic environments and multi-objective demands. The probabilistic application of these strategies works synergistically, enabling ISBOA to efficiently solve complex three-dimensional sensor node deployment optimization problems. For clarity, the process is shown in Figure 3.

#### 3.1.1. Initialization Phase

We denote the population of candidate solutions as $X={\left[X_{1},X_{2},\ldots ,X_{i},\ldots ,X_{\mathrm{SizePop}}\right]}^{T}$, where $X_{i}=\left[X_{i,1},X_{i,2},\ldots ,X_{i,\mathrm{Dim}}\right]$ represents the *i*-th candidate solution, and Dim denotes the dimensionality. Its position in the search space corresponds to the values of teh decision variables. Therefore, we randomly initialize the positions of the sensor nodes in the search space using Equation (Equation 12).(12)$$X_{i,j}=lb_{j}+\mathrm{rand}\times \left(ub_{j}-lb_{j}\right),\;i=1,2,\ldots ,\mathrm{SizePop},\;j=1,2,\ldots ,\mathrm{Dim}$$
where $lb_{j}$ and $ub_{j}$ represent the lower and upper bounds of the *j*-th dimension, respectively, and rand is a random number between 0 and 1.

Each $X_{i}$ represents a candidate solution. By substituting the candidate solution into the fitness function $\mathrm{FitFunc}(X_{i})$, the corresponding fitness value is obtained. The fitness value reflects the quality of the candidate solution.

In subsequent simulations, 23 benchmark functions are used to evaluate the performance of the swarm intelligence optimization algorithms, and the final solution is applied to the newly constructed multi-objective optimization problem.

In the field of optimization, 23 benchmark functions are commonly used to test the performance of the optimization algorithms. These functions have known global optimal solutions, which help evaluate the optimization ability, convergence speed, and robustness. They include five unimodal functions, six multimodal functions, seven other complex functions, and five high-dimensional and hybrid benchmark functions [25].

#### 3.1.2. Variation of Gaussian Cuckoo

Gaussian mutation is an effective optimization strategy that generates new solutions by applying random numbers, following a normal distribution, to the original position vector. When the random variable is close to the mean μ, the probability density value, or the offset, is large, which helps the algorithm break out of local optima, expand the search range, and explore potential global optima. On the other hand, when the random variable deviates from the mean μ, the offset is small, which aids in fine-tuning the local search and improves the algorithm’s optimization accuracy in local areas. At the same time, to fully consider the characteristics of the current population, two positions are randomly selected from the population. The difference between their positions interacts with the Gaussian distribution operator, forming the Gaussian mutation operator, denoted by *O*, as expressed by Equation (Equation 13).(13)$$O=G_{\mathrm{gauss}}\left(\zeta \right)\left(X_{\mathrm{rand}1}-X_{\mathrm{rand}2}\right),G_{\mathrm{gauss}}\left(\zeta \right)=\frac{1}{\sqrt{2\pi \sigma^{2}}}\exp\left(-\frac{{\left(\zeta -\mu \right)}^{2}}{2\sigma^{2}}\right)$$
where $G_{\mathrm{gauss}}\left(\zeta \right)$ represents the Gaussian distribution formed by the probability density, σ is the standard deviation, and μ is the mean or expected value of the distribution. Here, μ and σ are set to 0 and 1, respectively. To control the direction and magnitude of the mutation and balance exploration with exploitation, ζ is chosen between 0 and 1. $X_{\mathrm{rand}1}$ and $X_{\mathrm{rand}2}$ represent two randomly selected sensor node positions from the population.

Cuckoo search is a natural optimization algorithm inspired by the parasitic reproduction behavior of cuckoos. In this algorithm, nests represent potential solutions, and each nest corresponds to a solution. During iteration, cuckoos randomly select a nest and replace its current solution with a newly generated, better solution, thereby accelerating the search process. If a nest’s egg is discovered (i.e., the solution is not optimized), the nest is abandoned, and a new solution is randomly generated. The algorithm achieves a balance between generating new solutions randomly and selecting optimal solutions, combining global and local search capabilities. The discovery of a cuckoo egg is determined by $k_{\mathrm{IsFind}}$, as expressed by Equation (Equation 14).(14)$$k_{\mathrm{IsFind}}={\mathrm{Judge}}^{\mathrm{Dim}}>\xi$$
where ${\mathrm{Judge}}^{\mathrm{Dim}}$ is a random vector of dimension Dim, and ξ is the threshold for deciding whether to abandon the current nest.

When the owner of a bird’s nest discovers a cuckoo’s egg, the cuckoo will move randomly to the next position, which is considered a feasible area. The strategy behind the random movement of the cuckoo is expressed by Equation (Equation 15).(15)$$C=k_{\mathrm{IsFind}}\cdot LF\left(\mathrm{Dim}\right)$$

The position update of the Gaussian cuckoo mutation strategy is shown in Equation (Equation 16).(16)$$X_{i,j}^{\mathrm{new}}=X_{i,j}+O\cdot C$$

Finally, Equation (Equation 17) is used to determine whether to select the updated position.(17)$$X_{i}=\left\{\begin{array}{cc} X_{i}^{\mathrm{new}}, & \mathrm{if}\;\mathrm{FitFunc}\left(X_{i}^{\mathrm{new}}\right)<\mathrm{FitFunc}\left(X_{i}\right)\\ X_{i}, & \mathrm{otherwise} \end{array}\right.$$

The combination of the global search capability of Gaussian mutation and the fast convergence of cuckoo search enables the algorithm to search the solution space comprehensively, while precisely optimizing sensor node positions. This improves network coverage, connectivity, and efficiency. The integration of these two strategies avoids premature convergence to local optima, enhancing the algorithm’s stability and optimization performance, ensuring that wireless sensor networks exhibit greater adaptability in complex environments.

#### 3.1.3. Secretary Bird Predation Strategy

The main part of the ISBOA algorithm is entirely based on the SBOA, simulating the biological learning behavior of the secretary bird. The total time of the exploration phase *T* is divided into three equal parts. The three phases, denoted as $t\in [0,\frac{1}{3}T)$, $t\in [\frac{1}{3}T,\frac{2}{3}T)$, and $t\in [\frac{2}{3}T,T]$, are named as looking for prey, consuming prey, and attacking prey, respectively.

When $t\in [0,\frac{1}{3}T)$, it is similar to the initial iterations of an optimization algorithm, with a focus on global exploration. In this phase, a differential evolution strategy is adopted, where new solutions are generated based on the differences between individuals, increasing the diversity of the population and avoiding local optima, thus improving the chances of finding the global optimum. In wireless sensor network deployment, the differential evolution strategy can effectively search the entire network area, locate the best sensor nodes, and ensure that the initial deployment avoids local optima, optimizing network coverage and communication efficiency. The update of the sensor positions is shown by Equation (Equation 18).(18)$$\mathrm{While}\;t<\frac{1}{3}T,\;X_{i,j}^{\mathrm{new}}=X_{i,j}+\left(X_{\mathrm{rand}1}-X_{\mathrm{rand}2}\right)\times R1$$

Here, $X_{\mathrm{rand}1}$ and $X_{\mathrm{rand}2}$ represent random candidate solutions in the first-phase iteration, R1 is an array of random numbers in the range [0, 1] with dimensions 1×Dim, and $X_{i,j}^{\mathrm{new}}$ represents the new candidate solution generated after the random process. Whether to select the updated solution is determined by Equation (Equation 17) above.

When $t\in [\frac{1}{3}T,\frac{2}{3}T)$, the algorithm introduces the Brownian motion (RB) [18] while combining the individual’s historical best position $X_{\mathrm{best}}$ for local search. One of the characteristics of Brownian motion is that the movement of the individual is completely determined by small steps and disordered random fluctuations, which enhances the algorithm’s ability to explore locally, prevents individuals from prematurely falling into local optima, and accelerates convergence to the global optimum. At the same time, by combining $X_{\mathrm{best}}$ and Brownian motion, the sensor node positions are optimized in this phase, especially fine-tuning based on already good solutions, thus improving network coverage, connectivity, and controlling costs. The position update in this phase is expressed by Equation (Equation 19).(19)$$\mathrm{While}\;\frac{1}{3}T<t<\frac{2}{3}T,\;X_{i,j}^{\mathrm{new}}=X_{\mathrm{best}}+\exp\left({\left(\frac{t}{T}\right)}^{4}\right)\times \left(RB-0.5\right)\times \left(X_{\mathrm{best}}-X_{i,j}\right)$$

Finally, Equation (Equation 17) is used to determine whether to select the updated solution.

When $t\in [\frac{2}{3}T,T]$, this process introduces the Lévy flight strategy, which is a random motion model with both long and short step lengths. By randomly switching between large and small step lengths, it achieves a dynamic balance between global search and local optimization. This balance helps improve the algorithm’s convergence speed and prevents premature convergence to undesirable local optima. In addition, we introduce a nonlinear disturbance factor ${\left(1-\frac{t}{T}\right)}^{2\cdot \frac{t}{T}}$, which adjusts the search step length through gradually decreasing effects and adaptive search, thus achieving dynamic balance in the search process. The larger disturbances in the early stages help explore the solution space widely, while as time progresses, the disturbances decrease, and the algorithm focuses more on local optimization, avoiding premature convergence and ultimately accelerating convergence to the global optimum. This makes the algorithm more adaptive and flexible. These two strategies work together to not only help improve network coverage performance and connectivity but also optimize node deployment and reduce costs. The position update in this phase can be modeled by Equation (Equation 20).(20)$$\mathrm{While}\;t>\frac{2}{3}T,\;X_{i,j}^{\mathrm{new}}=X_{\mathrm{best}}+\left({\left(1-\frac{t}{T}\right)}^{2\times \frac{t}{T}}\right)\times X_{i,j}\times \frac{\mathrm{LF}(\mathrm{Dim})}{2}$$
where LF(Dim) represents the Lévy flight function, as expressed by Equation (Equation 21).(21)$$\mathrm{LF}\left(x\right)=s\times \frac{u\cdot \sigma_{\mathrm{levy}}}{{\left|v\right|}^{1/\beta }},\;\sigma_{\mathrm{levy}}={\left(\frac{\Gamma (1+\beta )\cdot \sin\left(\pi \beta /2\right)}{\Gamma \left(\left(1+\beta \right)/2\right)\cdot \beta \cdot 2^{(\beta -1)/2}}\right)}^{1/\beta }$$

Here, *u* and *v* are random values in the range [0, 1], β is a constant with a value of 1.5, *s* is typically a small fixed constant used to control the step size ratio, and Γ() represents the Gamma function.

Therefore, by combining these three stages, the algorithm achieves a balance between global search and local optimization, thereby enhancing its search capabilities. In the early stage, the differential evolution strategy increases population diversity and global exploration, avoiding the problem of local optima. In the middle stage, Brownian motion, combined with the individual’s historical best position, performs a fine local search, preventing premature convergence to a local optimum. In the later stage, Levy flight and a nonlinear perturbation factor are introduced, achieving a dynamic balance between global search and local optimization, which improves convergence speed and ensures the algorithm gradually fine-tunes toward the global optimum. In general, the strategies of these three stages complement each other and jointly drive the effective application of the algorithm in the deployment of wireless sensor networks, improving network coverage and connectivity and reducing costs.

#### 3.1.4. Secretary Bird Avoidance Strategies

This phase simulates the following two escape strategies adopted by the secretary bird when facing threats: $C_{1}$ and $C_{2}$. In the $C_{1}$ phase, Brownian motion and a random disturbance factor are introduced, focusing on enhancing the diversity of solutions to avoid premature convergence. In wireless sensor network deployment, this helps to prevent the node layout from being overly concentrated, improving network coverage and communication redundancy, while flexibly optimizing multiple design objectives to adapt to changing network demands and environmental conditions. In the $C_{2}$ phase, random disturbance $R_{2}$ and weighted differences *K*, etc., are introduced, focusing on improving global search capability and convergence speed, effectively avoiding local optima and quickly finding better solutions. In wireless sensor network optimization, by rapidly avoiding undesirable solution regions, it helps in finding a better configuration for sensor nodes, improving the algorithm’s optimization efficiency. These two strategies are simulated with equal probability in the SBOA algorithm, as expressed by Equation (Equation 22).(22)$$X_{i,j}^{\mathrm{new}}=\left\{\begin{array}{cc} C_{1}:X_{\mathrm{best}}+\left(2\times RB-1\right)\times \left(1-\frac{t}{T}\right), & \mathrm{if}\;\mathrm{rand}<0.5\\ C_{2}:X_{i,j}+R_{2}\times \left(X_{\mathrm{rand}}-K\times X_{i,j}\right), & \mathrm{otherwise} \end{array}\right.$$

Here, $R_{2}$ represents an array of normally distributed random numbers with dimensions 1×Dim; $X_{\mathrm{rand}}$ represents a random candidate solution; and *K* is a randomly chosen integer, either 1 or 2.

#### 3.1.5. Smooth Development System

The smooth exploitation mechanism combines unordered dimension sampling, random crossover, and sequential mutation, enabling flexible sampling and the adjustment of the dimension space to enhance the algorithm’s exploration capabilities. Unordered dimension sampling allows the algorithm to explore multiple dimensions effectively, avoiding local optima and improving the comprehensiveness of deployment solutions in wireless sensor networks [20].

Unordered dimension sampling is inspired by sampling theory, treating dimension selection as a sampling process, where the sampling rate determines the number of selected dimensions and a portion of the dimensions is randomly sampled while others remain static. By randomly selecting dimensions, the method reduces proximity to optimal dimensions, prevents the population from becoming overly concentrated, and broadens the search space. Additionally, it avoids premature convergence in specific dimensions, ensuring greater diversity in the search process and enabling the coverage of a wider potential solution space. The sampling rate calculation is expressed by Equation (Equation 23).(23)$${\mathrm{Rate}}_{\mathrm{sample}}=\left\lceil \max\left(\frac{t}{\mathrm{MaxIt}},\varepsilon_{1}\right)\cdot \mathrm{Dim}\right\rceil$$
where ${\mathrm{Rate}}_{\mathrm{sample}}$ is the sampling rate; ⌈⌉ rounds up to the nearest integer; MaxIt is the maximum number of iterations; and $\varepsilon_{1}$ adjusts the algorithm’s early exploration capability, which is set to 0.1 in this context.

Unordered dimensional sampling explores the search space through random crossover and sequential mutation to achieve more efficient global search. The main purpose of random crossover is to enhance the exploration ability, typically by randomly selecting an individual as a pivot. The radius of the exploration area is determined based on the distance between two random individuals. A larger radius helps break out of local optima, expanding the search range and obtaining global distribution information of the population, while a smaller radius indicates that the algorithm is gradually entering the exploitation phase, strengthening the focus on local search and adaptively balancing exploration and exploitation.

In the three-dimensional search space, new position vectors $X_{i,j}^{new}$ are generated through randomly selected individuals $X_{rand1},\;X_{rand2},\;$ and $X_{rand3}$, helping escape from the local optimum where the original position vector $X_{i,j}$ is trapped. This process is not merely a random selection of positions but instead expands the exploration range by maintaining the directional difference between $X_{rand2}$ and $X_{rand3}$. Specifically, the absolute difference between $X_{rand2}$ and $X_{rand3}$ forces the search in each dimension to evolve in the same direction. This mechanism helps reinforce the search in a particular direction, but if the difference is too large, it may lead to a decrease in population diversity, which could affect the global search ability and hinder the effective evolution of the search agents. The random crossover is expressed by Equation (Equation 24).(24)$$X_{i,j}^{\mathrm{rc}}=X_{\mathrm{rand}1}-\left(X_{\mathrm{rand}3}-X_{\mathrm{rand}2}\right)$$

Sequential mutation addresses the degradation problem of random crossover, avoiding excessive reduction in the exploration radius that can lead to local exploitation. By performing spatial mutation between adjacent individuals, it prevents the search from prematurely focusing on a specific region. Sequential mutation calculates the arithmetic mean of adjacent individuals’ spaces, avoiding the errors associated with harmonic and geometric means, thereby improving the stability and diversity of the search process. Sequential mutation is expressed by Equation (Equation 25).(25)$$X_{i,j}^{\mathrm{new}}=\frac{X_{i,j}^{\mathrm{rc}}+X_{i,j}}{2}$$

At the same time, the decision to update the position of the solution is determined based on Equation (Equation 17).

Finally, for clarity, we have organized and summarized the optimization process of the ISBOA algorithm, presenting it in pseudocode form, as shown in Algorithm 1. The core idea is to introduce the Gaussian cuckoo strategy and the smooth development mechanism at the beginning and end of the optimization framework of the original SBOA algorithm.
**Algorithm** **1** Pseudocode of the ISBOA  1:Initialize problem settings: Dim, ub, lb, PopSize (*N*), MaxIt (*T*), CurrIter (*t*)  2:Initialize the population randomly  3:**for** t=1 to *T* **do**  4:    Update Candidate Solution $x_{best}$  5:    **The Cossboo cuckoo mutated:**  6:    **for** i=1 to *N* **do**  7:        Compute the Gaussian mutation of the *i*th Candidate Solution using Equation (Equation 13)  8:        Calculate the cuckoo’s random move step length for the *i*th Candidate Solution using Equations (14) and (15).  9:        Calculate and update the new state of the *i*th Candidate Solution using Equations (16) and (17).10:    **end for**11:    **SBOA main part:**12:    **for** i=1 to *N* **do**13:        **if** $t<\frac{1}{3}T$ **then**14:           Calculate new status of the *i*th Candidate Solution using Equation (Equation 18)15:           Update the *i*th Candidate Solution using Equation (Equation 17)16:        **else if** $\frac{1}{3}T<t<\frac{2}{3}T$ **then**17:           Calculate new status of the *i*th Candidate Solution using Equation (Equation 19)18:           Update the *i*th Candidate Solution using Equation (Equation 17)19:        **else**20:           Calculate new status of the *i*th Candidate Solution using Equation (Equation 20)21:           Update the *i*th Candidate Solution using Equation (Equation 17)22:        **end if**23:    **end for**24:    **for** i=1 to *N* **do**25:        **if** r<0.5 **then**26:           Calculate new status of the *i*th Candidate Solution using $C_{1}$ in Equation (Equation 22)27:        **else**28:           Calculate new status of the *i*th Candidate Solution using $C_{2}$ in Equation (Equation 22)29:        **end if**30:        Update the *i*th Candidate Solution using Equation (Equation 17)31:    **end for**32:    **Smooth Exploration System:**33:    **for** i=1 to *N* **do**34:        Calculate the sampling rate of *i*th Candidate Solution using Equation (Equation 23)35:        Update the *i*th Candidate Solution via random crossover in Equation (Equation 24).36:        Update the *i*th Candidate Solution via sequence mutation in Equation (Equation 25).37:    **end for**38:    Save the best candidate solution so far39:**end for**40:Output: The best solution obtained by ISBOA for the given optimization problem **return** Best solution

### 3.2. Minimum Spanning Tree Domain Reduction Strategy

In our designed multi-objective optimization problem, deploying a certain number of sensor nodes within the monitoring area and adjusting their positions appropriately can construct a wireless sensor network with better performance. Meanwhile, the scale of the problem is mainly affected by the number of potential deployment positions for the sensor nodes. The more positions available, the larger the problem scale, and the computational complexity increases accordingly.

Since this problem belongs to the NP-hard category, we solve it using swarm intelligence optimization algorithms. To improve the solving efficiency, we introduce a set of movable network nodes, with the same number as the sensor nodes to be deployed, and set the target monitoring points within the monitoring area as fixed network nodes. Firstly, the swarm intelligence optimization algorithm iteratively adjusts the positions of the movable network nodes, allowing them to gradually form a shorter connected path within the detection area. During this process, the algorithm comprehensively considers the communication distance and connectivity between nodes to ensure that the formed path can connect all target monitoring points while minimizing the total communication cost. Once a suitable connected path is found, we retain this path and remove the movable network nodes, thereby limiting the deployment range of the sensor nodes to the area around this path. This approach effectively reduces potential deployment positions and decreases the problem scale. Although it may discard some better solutions, it significantly shortens the solving time while still ensuring high network performance, making it of great practical value [21].

To better understand the effectiveness of this strategy, we provide a more intuitive demonstration with the aid of diagrams. As shown in Figure 4, in a closed space of size 6 × 8 × 10, there are six target points and 10 movable network nodes that must be deployed to ensure that the connected path between them is as short as possible. Initially, the sensor node population is randomly deployed in the monitoring space, as shown in Figure 4a. Subsequently, we use the swarm intelligence optimization algorithm to iteratively adjust the positions of the sensor nodes, with the result shown in Figure 4b.

At this point, we deploy sensor nodes around the formed minimum spanning tree, as shown in Figure 5, where the blue positions indicate the potential deployment locations of the sensor nodes. We define $D_{\mathrm{round}}$ as the threshold for the maximum distance of sensor node deployment around the minimum spanning tree. That is, the location of the sensor nodes’ deployment must not exceed the given threshold $D_{\mathrm{round}}$, in terms of the shortest straight-line distance to the minimum spanning tree.

Finally, to better understand and apply the minimum spanning tree-based domain reduction strategy, we have organized and summarized the optimization process of movable network nodes. This process is presented in pseudocode form, as shown in Algorithm 2. The core idea is to iteratively adjust the positions of the movable network nodes, use the minimum spanning tree algorithm for the evaluation, and determine a shorter path connecting the target points and movable nodes. In addition, all potential deployment locations surrounding this optimized path are identified.
**Algorithm** **2** Pseudocode of the MST-Based System  1:**Initialize:** Problem settings including Target_Position, Ma, Mb, Mc, Dim, Dispersion, PopSize (*N*), MaxIt (*T*), CurrIter (*t*)  2:Randomly initialize the locations of the network node group  3:**for** t=1 to *T* **do**  4:    **for** i=1 to *N* **do**  5:        Update the position of the *i*th candidate solution through the movement rules of a swarm intelligence optimization algorithm  6:        Substitute the *i*th candidate solution into the minimum spanning tree algorithm to obtain the fitness value  7:    **end for**  8:    Update the network node group to determine $x_{\mathrm{best}}$  9:**end for**10:**Output:** All potential sensor node deployment locations around the shortest path **return** Best solution

## 4. Simulation Experiment

The previous chapter introduced the implementation strategy of the ISBOA. In this chapter, we first evaluate the convergence and accuracy of the ISBOA by using 23 benchmark test functions [25], comparing its performance with that of the SBOA, PSO, WOA, and NGO, which are other swarm intelligence optimization algorithms. Then, we apply the ISBOA to a multi-objective optimization problem proposed based on a practical application scenario, to assess whether its performance in real-world applications meets the expected outcomes.

### 4.1. Performance of the ISBOA and Comparison Algorithms on 23 Benchmark Functions

We compared the ISBOA with the SBOA, PSO, WOA, and NGO in 23 benchmark functions. The simulation results demonstrate that the ISBOA outperforms other swarm intelligence optimization algorithms in the vast majority of benchmark functions. According to the experimental results in Figure 6, it is clear that the ISBOA exhibits a faster convergence speed and higher solution precision in F1–F13. Although its advantages are not as prominent in F14–F23, and for some complex functions (e.g., F18, F19, F20), its performance is comparable to or slightly worse than other algorithms, the ISBOA still demonstrates the characteristic of rapidly reducing fitness values, showcasing excellent optimization efficiency. In general, the ISBOA outperforms the SBOA, PSO, WOA, and NGO in optimizing the 23 benchmark functions. Through improved search mechanisms and dynamic adjustment strategies, the ISBOA not only overcomes the shortcomings of other algorithms on complex problems, but it also maintains fast and stable global optimization capabilities.

### 4.2. Performance of the ISBOA and Comparison Algorithms on 23 Benchmark Functions

Figure 7 presents the performance comparison of various algorithms across the 23 benchmark functions. The box plots allow us to visually assess the performance of the ISBOA compared to the other algorithms. By aggregating the results of 30 iterations for each algorithm, it is evident that the ISBOA exhibits significant advantages in several aspects, including overall performance, stability, robustness, and the scarcity of outliers. Specifically, for most benchmark functions (such as F1, F5, F6, etc.), the box of the ISBOA is low and close to the optimal solution, with the median nearly always located near the lower bound, indicating strong optimization ability and the capacity to quickly approach the optimal solution. The short box height and concentrated result distribution, along with small performance fluctuations, suggest that the ISBOA is highly adaptive and can maintain stability despite changes in initial conditions or problem characteristics. For complex benchmark functions (such as F18, F19, etc.), ISBOA also demonstrates low fluctuation and stable performance, showcasing its excellent ability to handle complex, multi-modal problems. Moreover, the ISBOA has very few outliers, indicating a better adaptability to special cases and the effective avoidance of local optima.

In comparison, the original algorithm SBOA exhibits slightly worse overall stability than the ISBOA, occasionally showing outliers, as seen in functions F18 and F19. This may be due to the lack of improvements in the SBOA, such as better initialization or dynamic adjustment strategies, resulting in lower search efficiency and less stable result distribution compared to the ISBOA. NGO performs worse than the ISBOA in some benchmark functions (such as F13 and F16), with a higher median and greater fluctuation in the result, indicating insufficient optimization accuracy and stability. Its search mechanism seems to favor local exploitation, making it difficult to ensure global search capability for complex or high-dimensional problems. The WOA has a higher median in most benchmark functions, limited optimization accuracy, and greater fluctuation with significant outliers for complex functions. This may be due to an excessive local search during the spiral update and prey encirclement phases, weakening its global search ability. PSO shows a wider result distribution in several benchmark functions, with a higher box and significantly worse performance than the ISBOA. PSO tends to get stuck in local optima, making its performance unstable in high-dimensional or complex problems, and lacking sufficient search diversity, which affects its global exploration capability.

Overall, the ISBOA overcomes these shortcomings through improved initialization strategies, dynamic parameter adjustments, and a balance between global and local search, demonstrating superior performance and robustness.

### 4.3. The Performance of the ISBOA and the Comparison Algorithms in the New Multi-Objective Optimization Problem

To better understand the role of swarm intelligence optimization algorithms in this multi-objective optimization problem, we simulate the iterative process of deploying sensor node positions in a wireless sensor network (WSN) using a swarm intelligence optimization algorithm. In a monitoring space with dimensions 10×8×6, there are six target points that need to be monitored. Each target point includes its own position and type information, with a single target requiring specific sensor types for effective monitoring. This special target is colored red to distinguish it from other ordinary target points. This target point must be monitored by both T-type and H-type sensors simultaneously to be effectively monitored. That is, the target is considered to be effectively monitored only when both T-type and H-type sensors cover it, or when a (T+H) integrated type sensor covers it. In this scenario, 12 sensor nodes must be deployed for the best network performance. The number of sensor nodes selected is adequate to clearly demonstrate the process in the simulation experiment. The 12 sensor nodes are of the following types: two T-type single-function sensors, two H-type single-function sensors, two P-type single-function sensors, and six (T+H) integrated multifunction sensors. Only the special target point has specific sensor type coverage requirements, whereas all other ordinary target points are considered effectively covered as long as they are covered by any type of sensor.

Based on these conditions and constraints, the following requirements must be satisfied:The wireless sensor network must be fully connected;Each sensor must be able to communicate with at least two other sensor nodes, excluding itself;Each target point must be covered by at least two sensor nodes;Each potential location can host at most one sensor node;The specific target point must be covered by the specific sensor type, with a coverage degree of at least 2.

We use the ISBOA algorithm to iteratively optimize the positions of the 12 sensor nodes, enabling them to progressively satisfy all constraints and achieve better network performance. Figure 8a and Figure 8b show the positions of the sensor nodes before and after the iteration, respectively.

Initially, the fitness value of the wireless sensor network was 4.3333, with a coverage metric value of 1 and a connectivity metric value of 3.3333. Through the iterative optimization of the sensor node positions using the ISBOA algorithm, the fitness value of the wireless sensor network gradually increased to 10.3333, with a coverage metric of 3.667 and a connectivity metric of 6.667. Clearly, the coverage performance and connectivity performance of the heterogeneous wireless sensor network improved as the sensor node positions were iteratively optimized.

Figure 9, Figure 10 and Figure 11 illustrate the *XY*, *XZ*, and *YZ* cross-sections of the optimized heterogeneous wireless sensor network, respectively. These figures further help verify whether all the constraints of the newly proposed multi-objective optimization problem have been satisfied.

Meanwhile, we conducted experiments to compare the performance of the ISBOA with other benchmark algorithms in the new multi-objective optimization problem. The experimental results are shown in Figure 12. In the figure, the X-axis represents the number of sensor nodes, which is variable in the subsequent experiments and not fixed. The Y-axis represents the deployment cost of the sensor node population, with the deployment cost of each type of sensor node set according to the actual situation. In addition, the cost of integrated sensors formed by combining two or more single-function sensors is lower. The Z-axis represents the fitness value, which is the sum of coverage and connectivity. The figure shows the final fitness value of the WSN following iterative updates through various swarm intelligence optimization algorithms. As the number of sensor nodes and deployment costs increase, the performance of the WSN improves significantly. It should be noted that, regardless of the combination of sensor node quantity and deployment cost, the performance of the WSN after ISBOA iterations is consistently better than that of other comparison algorithms, demonstrating the unique advantages of the ISBOA in the optimization process.

## 5. Conclusions

In the cutting-edge research field of WSN deployment optimization, this paper, based on the latest research findings, redefines the multi-objective optimization problem to optimize the deployment strategy of heterogeneous wireless sensor networks. Our study not only considers the interdependence between network coverage, connectivity, and cost but also introduces an additional cost dimension to meet the monitoring needs of different functional sensors in specific areas. However, despite the progress made, there is still significant room for improvement in our wireless sensor network deployment optimization model. In addition to balancing coverage performance, connectivity, and deployment costs, we must also account for the degradation of the communication range and coverage due to the long-term use of sensor nodes. The degradation may be caused by factors such as hardware aging, solid-state battery characteristics (e.g., capacity decay), environmental changes (e.g., temperature, humidity), and energy consumption. These factors can affect the coverage range and communication capability of the nodes to varying degrees, ultimately reducing the network’s performance. Therefore, the optimization model should incorporate these degradation factors and adopt strategies such as network maintenance, node replacement, or energy replenishment to maintain the network’s stability and efficiency. Additionally, obstacles in the monitored space (such as buildings, terrain variations, etc.) can significantly impact node deployment. These obstacles can cause signal attenuation or obstruction, affecting both network connectivity and coverage. Hence, in actual deployment, besides node spacing and cost considerations, spatial modeling and analysis based on the environment should be integrated to dynamically adjust the deployment strategy, ensuring the robustness and reliability of the network. Future optimization solutions could further leverage dynamic adaptive algorithms and fault recovery mechanisms to ensure the network remains efficient and stable during long-term operation, thus providing better solutions for the deployment of wireless sensor networks in complex environments.

## Figures and Tables

**Figure 1 sensors-25-01366-f001:**
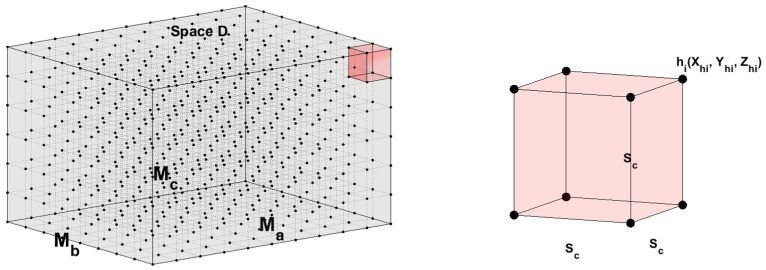
Monitoring scenario.

**Figure 2 sensors-25-01366-f002:**
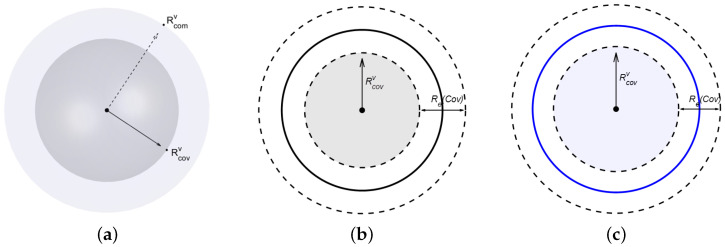
Coverage radius and communication radius of the sensor node. (**a**) Sensor coverage and communication radii. (**b**) Uncertainty in sensor coverage. (**c**) Uncertainty in sensor communication.

**Figure 3 sensors-25-01366-f003:**
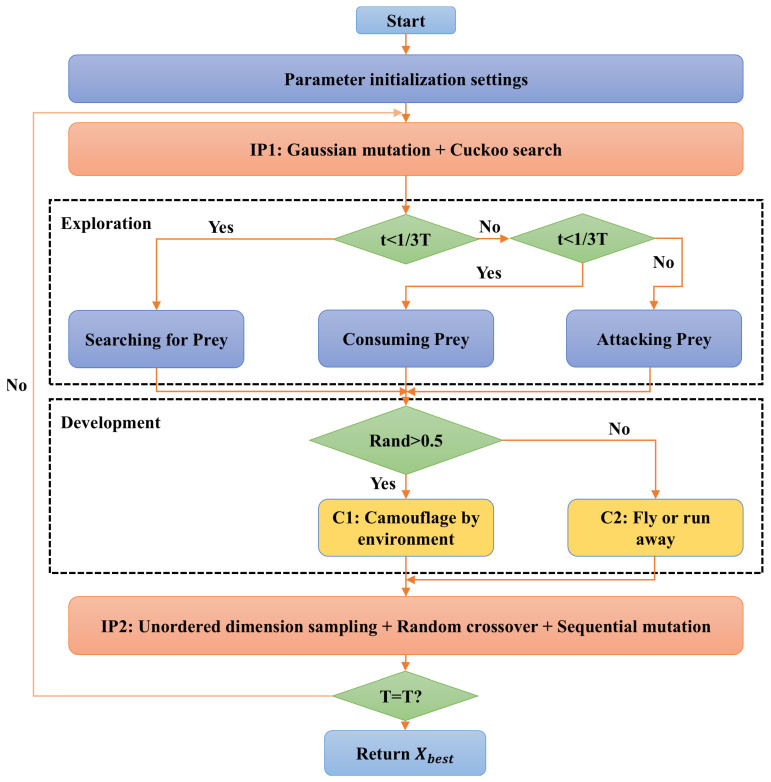
ISBOA flowchart.

**Figure 4 sensors-25-01366-f004:**
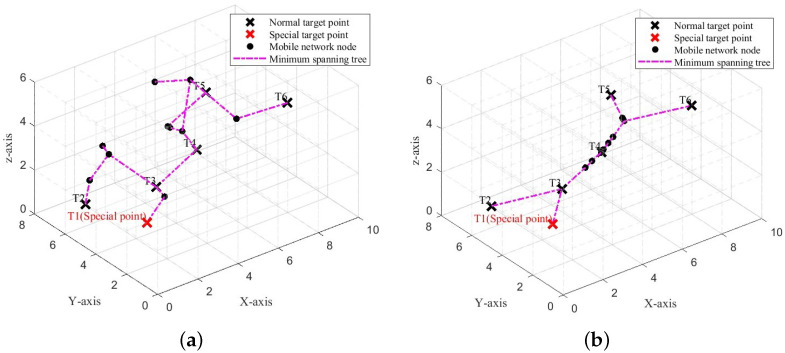
Position changes of movable network nodes in a closed space. (**a**) Initial deployment. (**b**) Optimized deployment.

**Figure 5 sensors-25-01366-f005:**
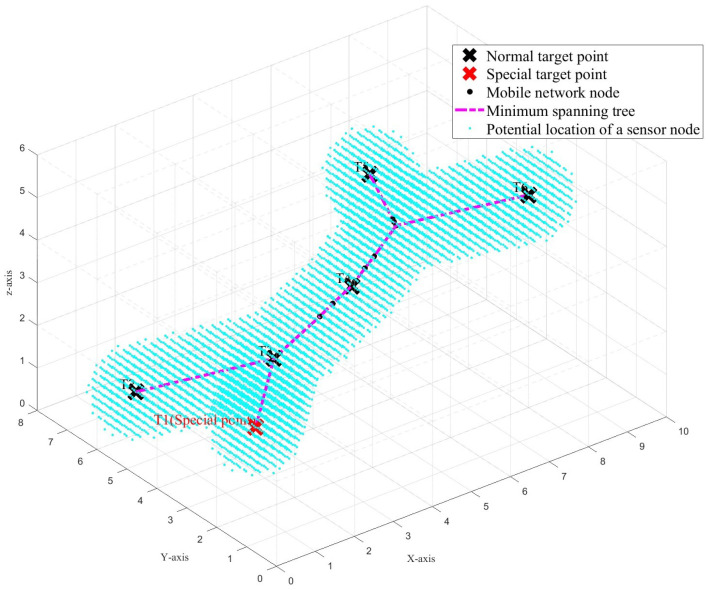
Potential deployment positions of sensor nodes.

**Figure 6 sensors-25-01366-f006:**
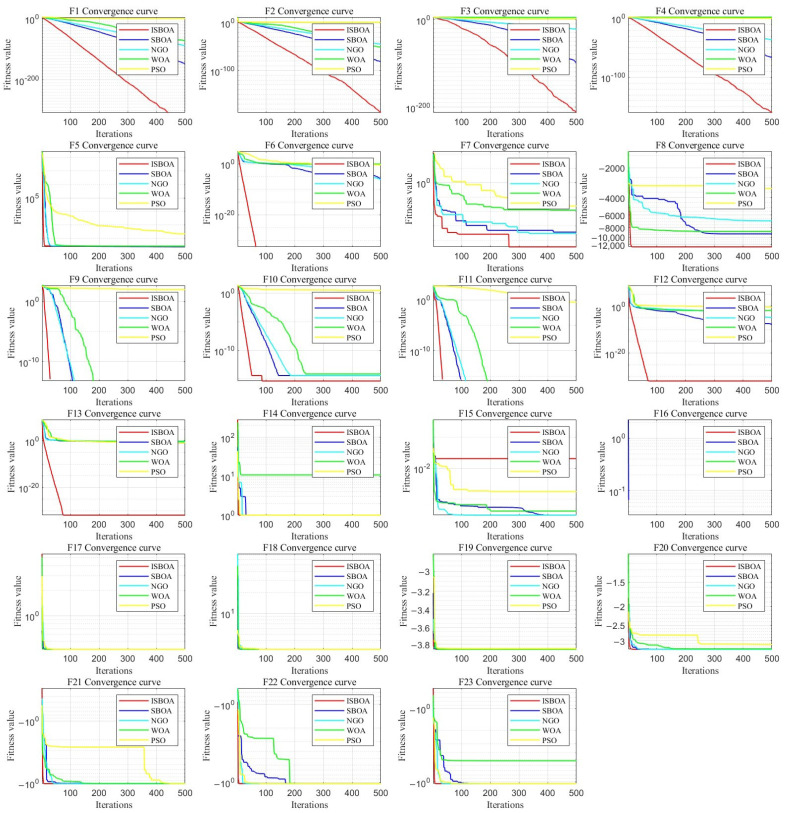
Potential Performance of ISBOA and comparison algorithms on 23 benchmark functions.

**Figure 7 sensors-25-01366-f007:**
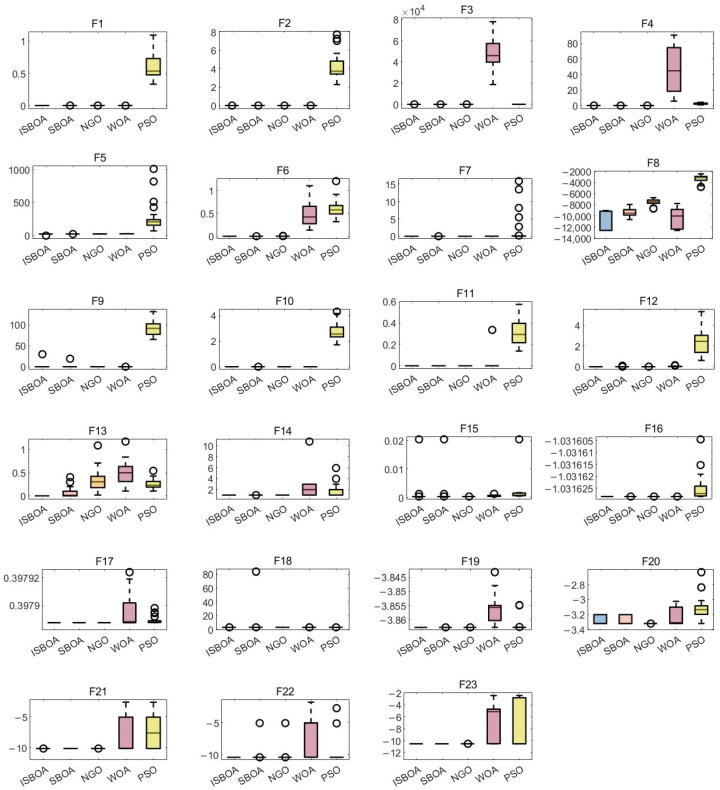
Performance Comparison of the ISBOA with other algorithms.

**Figure 8 sensors-25-01366-f008:**
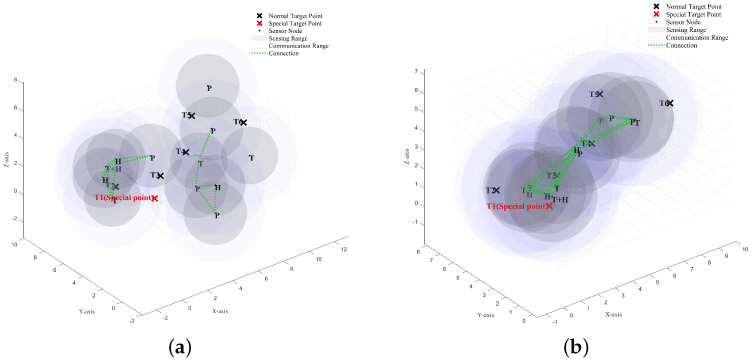
Sensor node positions before and after optimization using the ISBOA algorithm. (**a**) Initial positions of sensor nodes. (**b**) Optimized positions of sensor nodes.

**Figure 9 sensors-25-01366-f009:**
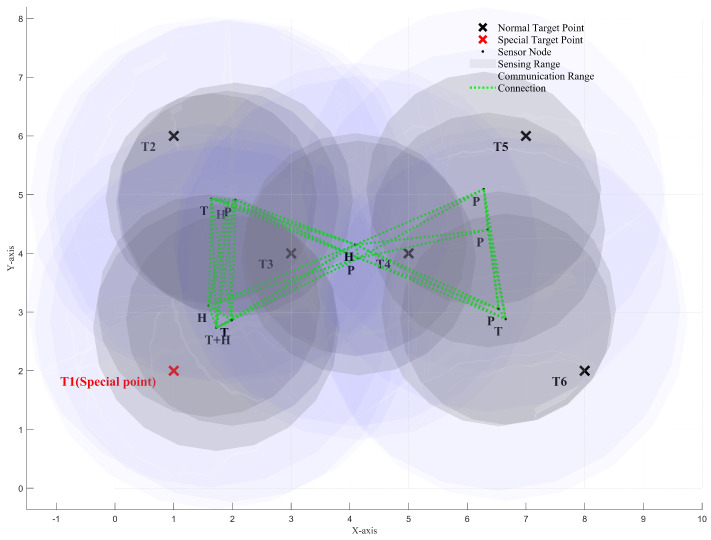
XY cross-section of the optimized heterogeneous wireless sensor network.

**Figure 10 sensors-25-01366-f010:**
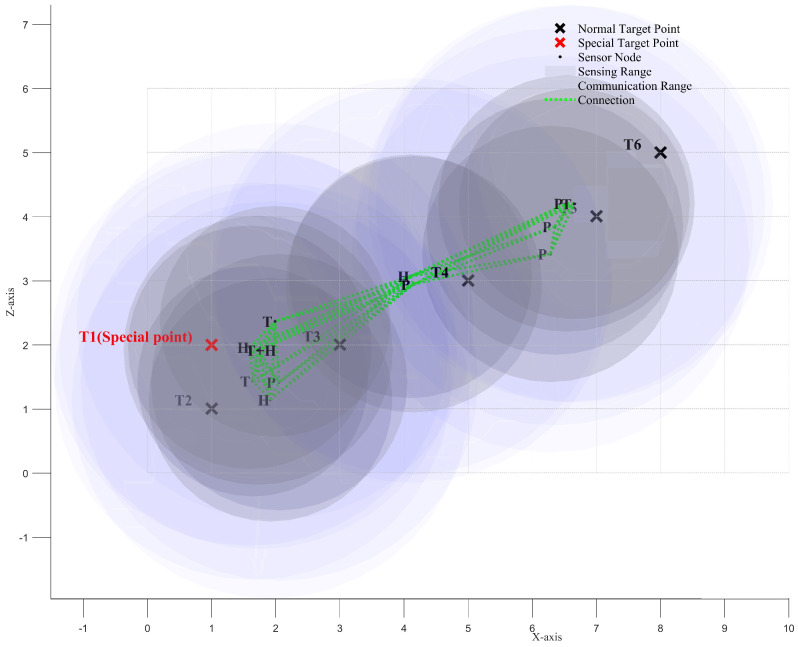
XZ cross-section of the optimized heterogeneous wireless sensor network.

**Figure 11 sensors-25-01366-f011:**
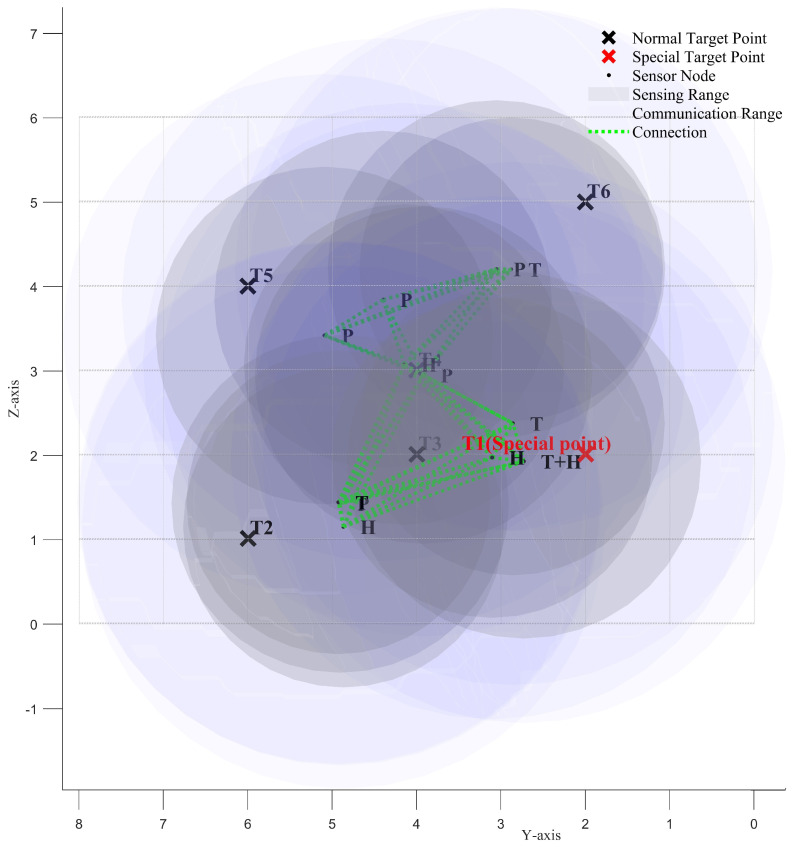
YZ cross-section of the optimized heterogeneous wireless sensor network.

**Figure 12 sensors-25-01366-f012:**
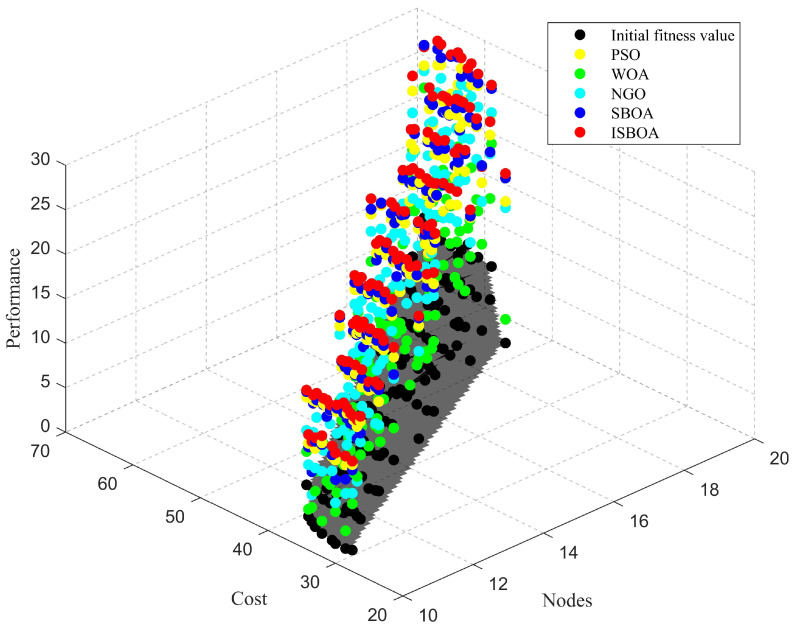
The performance of ISBOA compared to other algorithms in multi-objective optimization problems.

**Table 1 sensors-25-01366-t001:** Comparison Table for past, latest, and improved research.

	Past Research	Latest Research	My Improvement Points
Problem Design	Previous studies only focused on optimizing a single objective in WSN, namely, achieving high coverage with a limited number of sensor nodes.	Simultaneously balance coverage, connectivity, and deployment cost in WSN. While ensuring the performance of the WSN, the conditions of full network connectivity, only one sensor node per location, as well as the K-coverage of target points and C-connectivity between sensor nodes, are met.	Building upon the latest research and taking into account the integrable characteristics of sensor nodes, an additional dimension of the deployment costs has been introduced. Under the constraints of the latest research, considering the data inconsistency in heterogeneous functional wireless sensor networks, a constraint model is designed to address the monitoring requirements of different functional sensors in a specific area.
Algorithm Design	Used traditional swarm intelligence optimization algorithms (e.g., PSO, WOA) to solve NP-hard problems.	Appropriate improvements based on previous optimization algorithms enhance the solution accuracy and convergence speed in experiments.	Based on the latest swarm intelligence optimization algorithms, integrated Gaussian cuckoo bird mutation mechanism, and smoothing development mechanism to enhance solving capability.
Others	None	None	Proposed a minimum spanning tree domain reduction strategy to improve efficiency with minimal accuracy loss.

## Data Availability

The data are contained within the article.

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
