# Peer review of "3D Deployment Optimization of Wireless Sensor Networks for Heterogeneous Functional Nodes"

_sensors, 2025, doi:10.3390/s25051366_

Round 1

Reviewer 1 Report

Comments and Suggestions for Authors

This paper presents 3D deployment optimization of wireless sensor networks for heterogeneous functional nodes. Although the research is very interesting, however, I have the following concern to improve the quality of the paper:

1.      The present abstract did not acknowledge the effort of existing literature. I suggest the authors should highlight what has been done to address similar problems and how did their work differ from the existing knowledge.

2.      Secondly, in abstract the authors claimed to have achieved better performance in comparison to benchmarking work. I suggest that the convergence speed and higher accuracy value should be highlighted for readers to see the significant of their contribution.

3.      It unclear whether the authors combined background of the study with literature review section. Although, there is no acceptable standard for such structure. However, for better readability the former and the latter should be discussed separately. The present background section is too shallow for the readers to understand the need of the present study. I recommend the author should extensively discuss the background thereby highlighting the research gap.

4.      I suggest a comparison table at section 1.2 to indicate the difference between various related work and the proposed study

5.      Clearly highlight your research contribution at section 1.3 to avoid duplicate of effort

6.      Cite equation by their numbers instead of words. For example, as shown in in equation 1…

7.      The subheading e.g. 2.1 – 2.4 in section 2, may be confusing, there is need to briefly mention the four subheadings at the beginning of 2.0. explain the linkage between them and how they work. The authors may probably include a chart to illustrate the process

8.      What is the need for integrating ISBOA with Gaussian cuckoo mutation and smooth development mechanism. Both metaheuristic algorithms have shown remarkable performance for various optimization problems. The justification for the hybridization is unclear.

9.      Combining several metaheuristic algorithms can face additional computational complexity issues especially in wireless environments due to their dynamic nature. However, there is need for the authors to analyses the computation complexity of their solutions. This way, readers can understand the effectiveness of the proposed study.

10.  Algorithms 1 -3 were not properly explained and there is no in text citation of how they work. I Suggest a proper explanation while relating/citing any of the algorithms.

Comments on the Quality of English Language

The present English language can be acceptable, however, it could be improved for better readability.

Reviewer 2 Report

Comments and Suggestions for Authors

While the paper acknowledges factors like sensor node degradation and environmental changes, a deeper exploration into how these dynamic factors could be integrated into the optimization process could improve the robustness of the proposed solutions. Models that simulate the real-time behavior of nodes under various operational conditions would further strengthen the research.

Lack of Extensive Field Testing: The evaluation is largely simulation-based. While simulations provide substantial insights, incorporating real-life deployment cases or scenarios would allow for more comprehensive validation of the proposed methods and their real-world applicability.

The coverage and connectivity metrics are treated independently. In practice, these metrics may influence each other, and a more integrated approach could provide a holistic view of network performance.

Conclusions should include only main findings.

Reviewer 3 Report

Comments and Suggestions for Authors

This paper proposes a new multi-objective optimization problem aimed at optimizing the 3D deployment of wireless sensor networks to balance coverage, connectivity, and cost, while meeting the sensing requirements of specific areas. However, before further consideration, the following issues need to be addressed:

  1. The description of the ISBOA process, especially during the exploration phase and development phase, lacks detailed logical connections between the formulas and the process. In particular, the way Brownian motion and Levy flight strategies are combined to enhance search capabilities is not clearly defined.
  2. When discussing "Gaussian mutation," the formula does not clarify how the mutation amplitude and impact are calculated. Additionally, there is no detailed explanation of the mathematical principles behind Brownian motion, especially regarding how it adjusts from the historical position during the optimization process.
  3. In the communication model section, formula (6) defines communication probability, but it does not explain how these probabilities are used in practical applications to determine if a sensor can communicate effectively.
  4. The goal of the minimum spanning tree strategy is to improve efficiency by reducing the space for sensor location selection, but there is insufficient detail on how this strategy is implemented, particularly in large-scale optimization applications.
Comments on the Quality of English Language

The paper contains complex and lengthy sentences with unclear phrasing, lacking sufficient explanations of technical terms and concepts, which may hinder readability and understanding.

Round 2

Reviewer 1 Report

Comments and Suggestions for Authors

The authors have addressed most of my comments

Reviewer 3 Report

Comments and Suggestions for Authors

No further comments